# Whole Genome Sequencing of SARS-CoV-2: Adapting Illumina Protocols for Quick and Accurate Outbreak Investigation during a Pandemic

**DOI:** 10.3390/genes11080949

**Published:** 2020-08-17

**Authors:** Sureshnee Pillay, Jennifer Giandhari, Houriiyah Tegally, Eduan Wilkinson, Benjamin Chimukangara, Richard Lessells, Yunus Moosa, Stacey Mattison, Inbal Gazy, Maryam Fish, Lavanya Singh, Khulekani Sedwell Khanyile, James Emmanuel San, Vagner Fonseca, Marta Giovanetti, Luiz Carlos Alcantara, Tulio de Oliveira

**Affiliations:** 1KwaZulu-Natal Research Innovation and Sequencing Platform (KRISP), School of Laboratory Medicine & Medical Sciences, University of KwaZulu-Natal, Durban 4001, South Africa; pillaysureshnee@gmail.com (S.P.); jennifer.giandhari@gmail.com (J.G.); houriiyah.tegally@gmail.com (H.T.); ewilkinson83@gmail.com (E.W.); benjiechim@gmail.com (B.C.); LessellsR@ukzn.ac.za (R.L.); staceymattison@outlook.com (S.M.); inbal.gazy@mail.huji.ac.il (I.G.); maryam.fish@gmail.com (M.F.); Singhl@ukzn.ac.za (L.S.); kskhanyile@gmail.com (K.S.K.); sanemmanueljames@gmail.com (J.E.S.); vagner.fonseca@gmail.com (V.F.); 2Centre for AIDS Programme of Research in South Africa (CAPRISA), Durban 4001, South Africa; 3Department of Virology, National Health Laboratory Service, University of KwaZulu-Natal, Durban 4001, South Africa; 4Infectious Diseases Department, Nelson R Mandela School of Medicine, University of KwaZulu-Natal, Durban 4001, South Africa; Moosay@ukzn.ac.za; 5Laboratorio de Genetica Celular e Molecular, ICB, Universidade Federal de Minas Gerais, Belo Horizonte 31270-901, Brazil; luiz.alcantara@ioc.fiocruz.br; 6Laboratório de Flavivírus, Instituto Oswaldo Cruz Fiocruz, Rio de Janeiro 21045-900, Brazil; giovanetti.marta@gmail.com; 7Department of Global Health, University of Washington, Seattle, WA 98195, USA

**Keywords:** sequencing, SARS-CoV2, Illumina, protocols, COVID-19, bioinformatics

## Abstract

The COVID-19 pandemic has spread very fast around the world. A few days after the first detected case in South Africa, an infection started in a large hospital outbreak in Durban, KwaZulu-Natal (KZN). Phylogenetic analysis of severe acute respiratory syndrome coronavirus 2 (SARS-CoV-2) genomes can be used to trace the path of transmission within a hospital. It can also identify the source of the outbreak and provide lessons to improve infection prevention and control strategies. This manuscript outlines the obstacles encountered in order to genotype SARS-CoV-2 in near-real time during an urgent outbreak investigation. This included problems with the length of the original genotyping protocol, unavailability of reagents, and sample degradation and storage. Despite this, three different library preparation methods for Illumina sequencing were set up, and the hands-on library preparation time was decreased from twelve to three hours, which enabled the outbreak investigation to be completed in just a few weeks. Furthermore, the new protocols increased the success rate of sequencing whole viral genomes. A simple bioinformatics workflow for the assembly of high-quality genomes in near-real time was also fine-tuned. In order to allow other laboratories to learn from our experience, all of the library preparation and bioinformatics protocols are publicly available at protocols.io and distributed to other laboratories of the Network for Genomics Surveillance in South Africa (NGS-SA) consortium.

## 1. Introduction

In late December 2019, a mysterious viral pneumonia which had infected a number of people in Wuhan, China, was attributed to a new coronavirus [1]. This virus was labeled severe acute respiratory syndrome coronavirus 2 (SARS-CoV-2) and defined as the causal agent of Coronavirus Disease 2019 (COVID-19) [2]. Despite widespread attempts to contain the virus in China, within a few months, the outbreak had reached and affected 215 countries and territories around the world, including all countries in Africa.

Localized outbreaks are common when SARS-CoV-2 is introduced in a new geographic area. These outbreaks require urgent testing and tracing, identification of the causative agent, and epidemiological investigations to allow for appropriate infection control and for clusters of patients to be identified [3]. Genomic sequencing can be used to rapidly and accurately identify the transmission routes of the pathogen [4,5]. This can then be used to trace the path of transmission within a population and to possibly identify the probable source, potentially leading to an improved public health response [6,7,8,9,10,11,12].

On 5 March 2020, the first COVID-19 case was reported in KwaZulu-Natal, South Africa, from a traveler from Italy. On 9 March 2020, another returned traveler from Europe started a large chain of infection in a hospital in Durban, KwaZulu-Natal [13]. On 15 March 2020, South Africa declared a state of emergency, and on 27 March, a countrywide lockdown was implemented, which included grounding most of the international flights.

This study outlines the processes involved in setting up genomic sequencing in order to investigate the previously mentioned hospital outbreak in South Africa. The process was more complicated than expected, as, in addition to data needing to be generated quickly, problems were also encountered with unavailability of stock, importing sequencing reagents, and sample degradation and storage. However, we managed to set up two new protocols, which saved up to nine hours of hands-on time, as compared with the original ARTIC SARS-CoV-2 protocol.

## 2. Materials and Methods

### 2.1. Sample Collection and Preparation

Remnant samples from various diagnostic laboratories were obtained. These were collected by using nasopharyngeal and oropharyngeal swabs from qPCR-confirmed COVID-19 patients. These samples consisted of either the primary swab sample or extracted RNA. Inactivated swab samples were heated in a water bath at 60 °C for 30 min, prior to RNA extraction. RNA was extracted on an automated Chemagic 360 instrument, using the CMG-1049 kit (Perkin Elmer, Hamburg, Germany), or by manual extraction, using the QIAamp Viral RNA Mini Kit (Qiagen, Valencia, CA, USA). The RNA was stored at −80 °C prior to use.

The study was conducted in accordance with the Declaration of Helsinki, and the project was approved by the Biomedical Research Ethics Committee of the University of KwaZulu-Natal, South Africa, with protocol reference number BREC/00001195/2020.

### 2.2. Real-Time Polymerase Chain Reaction

RNA samples were thawed to room temperature prior to real-time PCR testing. They were tested for three SARS-CoV-2 genes, i.e., *ORF1ab*, *S protein*, and *N protein*, using the TaqMan 2019-nCoV Assay Kit v1 (LifeTechnologies, Frederick, MD, USA) according to the manufacturer’s instructions. In summary, a 20 µL master mix was prepared for each target gene (i.e., *ORF1ab*, *S protein*, and *N protein*) containing 11.25 µL of nuclease free water, 6.25 µL of TaqMan Fast Virus 1-Step Master Mix (4X), 1.25 µL RNase P (20X), and 1.25 µL of the 2019-nCoV target (20X), in respective tubes. Next, 20 µL of the master mix was added to a 96-well plate and 5 µL of RNA to the respective wells. A positive control (i.e., 1 µL TaqMan 2019-nCoV Control Kit v1 and 4 µL of nuclease free water) was included and no template control (i.e., 5 µL nuclease-free water) for each target gene.

Then, qPCR was performed on a QuantStudio 7 Flex Real-Time PCR instrument (Life Technologies), using the following conditions: 50 °C for five minutes, 95 °C for 20 s, 40 cycles of 95 °C for three seconds, and 60 °C for 30 s. Cycle thresholds (Cts) were analyzed by using auto-analysis settings, with the threshold lines falling within the exponential phase of the fluorescence curves and above any background signal. To accept the results, a Ct value for RNAse P was used as an endogenous internal amplification control in each reaction, and the results were only valid if the Ct values in the no-template control were undetermined.

### 2.3. Tiling-Based Polymerase Chain Reaction

The complementary DNA synthesis was performed by using SuperScript IV reverse transcriptase (Life Technologies) and random hexamer primers, followed by gene-specific multiplex PCR, using the ARTIC protocol, as described previously [13]. In summary, SARS-CoV-2 whole-genome amplification by multiplex PCR was attempted by using primers designed on Primal Scheme (http://primal.zibraproject.org/) to generate 400 base pair (bp) amplicons with 70 bp overlaps, covering the 30 kilobase SARS-CoV-2 genome. PCR products were purified in a 1:1 ratio, using AmpureXP purification beads (Beckman Coulter, High Wycombe, UK), and were quantified the purified amplicon, using the Qubit double strand DNA (dsDNA) High Sensitivity assay kit on a Qubit 4.0 instrument (Life Technologies). Amplicon fragment sizes were estimated on a LabChip GX Touch (Perkin Elmer) prior to library preparation.

### 2.4. Library Preparation and Next-Generation Sequencing

Depending on the availability of reagents, library preparation was attempted, using three different kits, namely TruSeq DNA Library Prep kit, NEBNEXT Ultra II DNA Library Prep Kit, and the Nextera DNA Flex Library Prep kit.

#### 2.4.1. TruSeq Nano DNA Library Preparation

The TruSeq Nano DNA library preparation kits (Illumina, San Diego, CA, USA) were used to prepare uniquely indexed paired-end libraries of genomic DNA, according to the manufacturer’s instructions. In summary, 200 ng of input DNA from purified amplicons was used for end-repair reaction, or 10 µL of neat sample in cases of insufficient concentration. Adapter ligation was used by using Illumina TruSeq Single DNA Indexes Set A and Set B (Illumina), and adaptor enrichment was performed by using eight cycles of PCR. The libraries were quantified by using the Qubit dsDNA High Sensitivity assay kit on a Qubit 4.0 instrument (Life Technologies), and the fragment sizes were analyzed by using a LabChip GX Touch (Perkin Elmer). Each sample library was normalized to 4 nM concentration, and the normalized libraries were pooled and denatured with 5 µL of 0.2 N sodium acetate. The 12 pM library was spiked with 1% PhiX control (PhiX Control v3) and sequenced on an Illumina MiSeq platform (Illumina), using a MiSeq Nano Reagent Kit v2 (500 cycles). This method could not be used further, as insufficient reagents were available for the library preparation.

#### 2.4.2. NEBNext Ultra II DNA Library Preparation

The NEBNext Ultra II DNA Library Preparation kits were used according to the manufacturer’s instructions. In summary, purified tiling PCR amplicons were diluted to 5 ng/µL, and End Prep reactions were performed on the amplicons, using NEBNext Ultra II End Repair/dA-Tailing Module (New England BioLabs, Ipswich, MA, USA). Due to the national lockdown, there was unavailability of stock of the NEBNext Multiplex Oligos from Illumina. As a result, adapter ligation was performed by using the remaining Illumina TruSeq Single DNA Indexes Set A and Set B (Illumina), and adapter enrichment was performed by using the remaining PCR Primer Cocktail from the Illumina TruSeq Nano DNA Library Prep kit. PCR cycling conditions were adapted accordingly for the reagent substitution, and eight cycles were used for the enrichment step. Size selection was performed by using 0.9× AmpureXP purification beads (Beckman Coulter), and the libraries were quantified by using the Qubit dsDNA High Sensitivity assay kit on a Qubit 4.0 instrument (Life Technologies). The fragment sizes were analyzed by using a LabChip GX Touch (Perkin Elmer), with expected fragments between 300 and 600 bp in size. Each sample library was normalized to 4 nM concentration, and the normalized libraries were pooled and denatured with 5 µL of 0.2 N sodium acetate. Then, 1% PhiX control (PhiX Control v3) was spiked in a 12 pM library and sequenced on an Illumina MiSeq platform (Illumina) using a MiSeq Nano Reagent Kit v2 (500 cycles).

#### 2.4.3. Nextera DNA Flex Library Preparation

The Nextera DNA Flex Library Prep kits (Illumina) were used according to the manufacturer’s instructions. Undiluted tiling PCR amplicons were used. Briefly, the DNA was tagmented with bead-linked transposomes, and the tagmentation reaction was stopped with tagmentation stop buffer before proceeding to the post-tagmentation cleanup, using the tagmentation wash buffer. This step was followed by eight cycles of amplification of the tagmented DNA with enhanced PCR mix and index adapters. The Nextera DNA CD indexes were used (Illumina). Then libraries were cleaned using 0.9x sample purification beads and eluted in 32 μL resuspension buffer. Libraries were quantified by using the Qubit dsDNA High Sensitivity assay kit on a Qubit 4.0 instrument (Life Technologies). The fragment sizes were analyzed by using a LabChip GX Touch (Perkin Elmer), with expected fragments between 500 and 600 bp in size. Each sample library was normalized to 4 nM concentration, and the libraries were normalized and denatured with 5 µL of 0.2 N sodium acetate. The 12 pM library was spiked with 1% PhiX control (PhiX Control v3) and sequenced on an Illumina MiSeq platform (Illumina), using a MiSeq Nano Reagent Kit v2 (500 cycles). All primers are provided in Appendix A.

This protocol is available at protocols.io coronavirus-method-development community website (https://www.protocols.io/view/illumina-nextera-dna-flex-library-construction-and-bhjgj4jw) since 17 June 2020.

### 2.5. Data Analysis

Raw reads from Illumina sequencing were assembled by using Genome Detective 1.126 (https://www.genomedetective.com/) and the coronavirus typing tool [14,15]. Genome detective preprocesses reads by using Trimmomatic to remove adapter and primer sites. Reads are then filtered and sorted, on a protein level, into “bins” corresponding to recognized viral species, using DIAMOND [16] and the UniRef90 database (https://www.uniprot.org/help/uniref). Reads in each bin are assembled by using SPADes assembler [17]. Genome detective produces consensus sequences from the assembled contigs for the identified viruses. In our SARS-CoV-2 assembly workflow, the initial assembly obtained from Genome Detective was polished by aligning mapped reads to the reference (NC_045512) and filtering out mutations with low genotype likelihoods, using bcftools 1.7-2 mpileup method. All mutations were confirmed visually, with bam files, using Geneious Prime software suite (Biomatters Ltd., Auckland, New Zealand) software, and the sequences were manually edited as required. The complete protocol, including heuristics to distinguish and edit wrong variant calls, can be accessed on our website and on protocols.io (https://www.krisp.org.za/ngs-sa/ngs-sa_genomic_and_bioinformatic_protocols_to_sars-cov-2/token/3). For samples with repeat sequencing, forward and reverse reads from all sequencing runs were merged, respectively, and assembled as one. All of the sequences were deposited in GISAID (https://www.gisaid.org/) (Accession IDs in Appendix A). Lineage assignments were established by using a dynamic lineage classification method proposed by Rambaut et al. [18] via the Phylogenetic Assignment of named Global Outbreak LINeages (PANGOLIN) software suite (https://github.com/hCoV-2019/pangolin). Moreover, 10,959 GISAID reference genomes (all authors acknowledged in Appendix A) and 54 whole genomes generated at KwaZulu-Natal Research Innovation and Sequencing Platform (KRISP) were aligned in Mafft v7·313 (FF-NS-2) [19], followed by manual inspection and editing in the Geneious Prime software suite (Biomatters Ltd., New Zealand). A maximum likelihood (ML) tree topology was constructed in IQ-TREE (GTR+G+I, no support) [20,21]. The resulting phylogeny was viewed and annotated in FigTree v1.4.4 (https://github.com/rambaut/figtree/releases) and R package ggtree v.1.4.11 [22]. All of the data produced have been deposited in the GISAID (consensus genomes) (Accession IDs in Appendix A) and at the FASTQ short reads deposited at the Short Read Archive (SRA) with accession code PRJNA636748.

## 3. Results

### 3.1. Sample Characteristics

A total of 108 SARS-CoV-2 (the virus) positive samples (sampled from the end of March to the beginning of May 2020) were received. Of these, 77 were from nasopharyngeal and oropharyngeal swabs, and 31 were extracted RNA. The average age of the patients was 51 years (ranging from 23 to 91 years), with a gender distribution of 70.5% females and 29.5% males (Appendix A). Of the 108 individuals, 63 were from the hospital outbreak, and 30 were from randomly selected positive individuals sampled in the same city but unrelated to the outbreak at the time of sampling. These last samples were used as control for the outbreak investigation. In addition, 13 samples delivered from the hospital outbreak were sequenced, but the patient or healthcare worker from the sample ID, or the sample ID, has been lost, or the sample itself was lost during transport or in storage, before coming to the laboratories.

### 3.2. SARS-CoV-2 Sequencing with Limited Reagents and Unavailability of Stock during a Pandemic in a Country in Lockdown

Sequencing commenced the day after this outbreak investigation started, on 5 April 2020, so there was no time to prepare. Priority was given to genome sequencing rather than qPCR, as there were limited sample volumes, and results were required urgently for outbreak investigations. One of the 108 samples failed library preparation; the remaining 107 samples were sequenced and used in the final analysis. It is important to note that these samples did not arrive at the same time. There were 12 shipments, and the quality of the samples was not optimal, as some were sent to multiple laboratories for diagnostics and were not well stored, i.e., many were stored for days or weeks at room temperature. Furthermore, an undisclosed number of samples were heat-inactivated, sometimes more than once.

Initially, the ARTIC protocol was used with no modification; this included the TruSeq library preparation step. However, this was a very lengthy and laborious process, which took close to twelve hours of hands-on time to produce the libraries for sequencing. Furthermore, reagents for only 24 samples were available, as the order of Illumina TruSeq DNA Library preparation kit (x 96 sample libraries), which was placed in February and was enough to produce 480 genomes, had not arrived, due to the restrictions on international flights. In order to complete the outbreak investigation, it was necessary to improvise and look for other reagents that could replace this library preparation kit and that were already available in South Africa.

Colleagues from six genomics laboratories in South Africa were approached, but, unfortunately, they could not provide a suitable library kit. Many companies in South Africa were contacted telephonically, to locate local stock. Two NEB Next Ultra II library kits were found at a local company, Inqaba Biotech. The kit was couriered from Johannesburg by car on 10 April. Unfortunately, one of the components, the NEBNext Multiplex Oligo, which was necessary to run the NEB Next Ultra II library kits, was missing. To overcome this, the protocol had to be changed to perform adapter ligation by using remaining Illumina TrueSeq Single DNA Indexes Set A and Set B. This process also involved adapting PCR conditions and skipping some steps of the original NEBNext Ultra II library protocol. Fortunately, this strategy was successful and produced good-quality fragment sizes, with expected fragments between 300 and 600 bp in size.

Illumina was also approached to chase up our TrueSeq order. However, Illumina informed us that they had recently changed distributors in South Africa and that they could not fulfil the previous order. At the time of writing this manuscript, 12 weeks after the order, the TrueSeq reagents have still not arrived. However, via the new distributors in South Africa, Separations Pty., Illumina has provided us with two large complementary library kits, the TruSeq and the Nextera Flex. The Nextera Flex library preparation kit was suggested by their technical team as a potentially better and quicker solution to produce SARS-CoV-2 genomes, which we found to be true.

Normally, a qPCR report is produced before sequencing is attempted. However, due to the limited RNA and urgent need to generate data to solve the nosocomial outbreak, a qPCR was generated in parallel to the sequencing process and only after genomes had been generated. In summary, it took three days to complete one round of sequencing, qPCR, and analysis of the data (Figure 1) in the laboratory. The sequencing process involves RNA extraction, RT-PCR, PCR amplification, library preparation, and Illumina sequencing, followed by genome assembly and sequence analysis. Twelve samples were run per round, and this was increased to 24 per round during the process. In total, 108 COVID-19 positive individuals were sequenced in less than three weeks.

In total, 102 of 108 (94.4%) samples had remaining RNA for qPCR. Of the 102 samples, 97 had positive Ct values for all three target genes, and five had at least one undetermined target gene. The median Ct value for the three genes targeted was 24.9 (21.4–29.8), with good agreement between the different probes, with mean Ct values of 26.5 (22.2–31.1), 25.2 (21.4–29.9), and 24.0 (20.8–30.2) for *ORF1ab*, *S protein*, and *N protein*, respectively.

### 3.3. SARS-CoV-2 Genome Assembly

The generation of high-quality genomes from the sequencing data was performed by using a three-step assembly and clean-up workflow (Figure 2). While the goal of genotype likelihood calculations was to improve the quality of variants called, it also helped to increase coverage along the length of the genomes, by optimizing the alignment of reads. Our assembly workflow involves a last manual step, when whole genomes are polished and all of the mutations are visually checked from bam alignments for confirmation. Some of the adaptors did not appear to be well filtered in the assembly process (potentially due to an older version of primers) and resulted in miscalled mutations. However, this was easily detected in the BAM files, and all the mutations were visually checked, and the sequences were manually edited as required. On average, consensus started with seven (4–13) mutations and ended up with five (4–6). The quality of the consensus sequences clearly increased, as indicated by significantly higher (*t*-test) coverage, concordance, identity, and matches at Step 2, compared to Step 1, and significantly higher matches at Step 3, compared to Step 2 (after visual validation) (Appendix A and Appendix A). The second step increased coverage because we are now mapping reads to a reference sequence, using mpileup and bcftools, while in the first step, it was a de novo assembly, using SPADES. De novo assembly will reject more reads than a reference-based assembly. This process produced 107 SARS-CoV-2 genomes, 54 of which were whole genomes (>90% coverage) and 53 partial genomes (mean genome length: 60.5%, IQR 51.5–72.4%).

### 3.4. Association between Ct Value and Genome Coverage in an Outbreak

There was a marked linear association between higher coverage and samples with a lower Ct value (R^2^ = 0.50) (Figure 3). There was a clear trade-off between Ct value and genome coverage, and three cutoff values were used to analyse the data, namely Ct < 25, <27, and <30 (Figure 3). Genomes of significantly higher coverage generated from samples with mean Ct values below the chosen thresholds are consistently shown. For example, Ct values were obtained for 51 of the 54 near-full-length genomes with a mean Ct value of 22.2 (19.7–24.2). For this outbreak, a Ct < 27 was shown to be a good threshold to obtain high-quality near-complete genomes. However, 12 samples with a Ct < 27 did not produce whole genomes (Figure 3). This can probably be attributed to sample degradation, as many of the samples have been stored at room temperature. An analysis of depth of coverage of the sequences in relation to Ct values is provided in Appendix A.

### 3.5. Comparison of Library Preparation Methods

The three different library preparation methods, i.e., TruSeq DNA, NEBNext Ultra II, and Nextera DNA Flex, produced similar results. In order to properly evaluate the performance of the library methods, a subset of multiple sequencing for 16 samples was generated, four of which were sequenced in triplicate and four in duplicate for each of the two new library preparation methods (Table 1). For example, ten of the 16 sequences produced with the different methods provided very similar genome coverage and mutations. In addition, using a Ct lower than 30, the TruSeq DNA and NEBNext Ultra II protocols produced 14 and 40 whole genomes, which represented 25.9% and 74.1% of the genomes produced, respectively.

The comparison of cost and hands-on time initially showed that the NEBnext Ultra II provided the most cost-effective solution, as all components cost approximately US $42 per genome, and the hands-on time was six hours. The Nextera Flex is the easiest and quickest method, with three hours of hands-on time, but it costs approximately US $60 per genome at full-reaction usage. The hands-on time of TruSeq, the original library in the ARTIC protocol, was twelve hours, and the cost for this study was US $52 per reaction. It is important to note that prices in South Africa are higher than most regions in the world and are deeply affected by currency fluctuations. In addition, some kits require additional components that need to be purchased. The Illumina Nextera DNA Flex and the TruSeq Nano DNA Library Preparation kits contain all the reagents required to perform the library preparation. The NEBNext Ultra II DNA kit does not include some of the reagents required for library preparation, such as the end-repair reagents, NEBNext Ultra II End Repair/dA-Tailing Module (New England BioLabs), and Ampure beads (Beckman Coulter). However, if the cost of the personnel and the extra reagents needed for the NEBNext Ultra II are taken into account, the Nextera Flex seems to be the most cost-effective and easy-to-use library preparation kit.

### 3.6. Implementation of Nextera Flex and Improvement of Process and Success Rate of Sequencing

The Nextera Flex library preparation method was optimized with samples with low and high Ct values. This optimized protocol was used to sequence an additional 66 SARS-CoV-2 samples from the population around the outbreak. The new protocol increased the success rate of whole genome sequencing to 90.9%, i.e., 60 whole genomes from 66 samples (Figure 4). This protocol was distributed in the protocol.io coronavirus-method-development community and by the time of writing this article has been accessed over 1000 times. 

### 3.7. Phylogenetic and Lineage Analysis for Near Full-Length Genomes

A very basic phylogenetic analysis of the near-full-length genomes was performed, as this analysis is beyond the scope of this paper. The first 54 near-full-length genomes belong to lineages B (*n* = 3), B.1 (*n* = 50), and B.2 (*n* = 1) (Figure 5, Appendix A). All of the outbreak sequences clustered closely to each other in the lineage B.1., and they were 99.99% identical, with one or two mutations that differentiated themselves. The number of mutations (mean of five mutations from the original Wuhan reference sequences) of most of the sequences were in line with other public sequences sampled at the same time. Furthermore, the outbreak sequences had a specific mutation, 16,736 C to T (Appendix A), which caused a non-synonymous mutation at the *ORFab1* gene position. A more detailed phylogenetic analysis is described in the dedicated early genomic epidemiology report of SARS-CoV-2 infection in KwaZulu-Natal (KZN) and the investigation of the hospital outbreak [6,23].

## 4. Discussion

Whole-genome sequencing of SARS-CoV-2 samples has the capacity to provide rapid and accurate information for the analysis of genetic variability and identification of transmission chains [24]. However, as outbreaks happen sporadically and cannot be predicted, it is not always possible to have all the resources required to perform the necessary tests in resource-limited settings. As in many countries, there was a national lockdown in South Africa during the SARS-CoV-2 pandemic, with limited flights in and out of the country, thus making it difficult for suppliers to deliver reagents. In addition, reagents for SARS-CoV-2 were in great demand in the world, making them more difficult to access.

The KRISP laboratory received training in Oxford Nanopore Technologies (ONT) SARS-CoV-2 sequencing from colleagues from Brazil, and the first two sequencing runs involved 24 samples (*n* = 12 each run) that were sequenced on 7 April 2020. However, the flow cells were over two years old and had less than 600 active pores each, which produced low-quality and lower-coverage genomes. Given the ONT problem, the ARTIC protocol [13] on the Illumina Miseq sequencing platform was used from 8 April 2020.

All samples from the outbreak investigation had initially been collected and tested at different private and government laboratories. Results were required urgently for genomic sequencing, and therefore tracing the exact qPCR results from the original laboratories to determine the Ct values was impractical. Ideally, as shown in other reports of SARS-CoV-2 genomic epidemiology [8], samples should be tested on qPCR first. Those with a Ct below 30 were shown to produce longer and higher-quality genomes. A recent study of high-throughput sequencing in Iceland showed up to a 90% success rate of sequencing SARS-CoV-2 near-full-length genomes for samples with Ct < 30. However, this was a prospective study in a very developed country. A Ct value of <27 provided the best cutoff to produce near-full-length genomes in this setting, using samples that had not been optimally collected.

In this study, the ARTIC Illumina method [13] was initially used to amplify the SARS-CoV-2, using a tiling PCR approach. This method recommends the use of the TruSeq DNA Nano kit for Illumina sequencing, which is a very labor-intensive approach (up to twelve hours of hands-on time). As reagents for TruSeq ran out and other methods needed to be set up quickly, a number of much better library sequencing kits that were either cheaper or were much less laborious were identified. The NEBnext Ultra II (New England BioLabs) and Nextera DNA Flex (Illumina) were evaluated and adopted. The NEBnext kit was used to generate the majority of the 108 genomes and 54 near-full-length genomes. The Nextera Flex DNA library preparation kit was also evaluated, and we found it saved up to nine hours of hands-on time, when compared with the original ARTIC protocol that uses TruSeq.

All three library preparation methods produced high-quality genomes. There was no significant difference in coverage. The marginally lower coverage observed with the NEBnext Ultra II kit could be due to the unavailability of the recommended adapters and the substitution with the TruSeq adapters. Slightly lower coverage seen with Nextera Flex DNA could be due to the samples being subjected to freeze–thaw cycles, as the Nextera Flex DNA kit was used a month after samples were received and tested using the other two kits.

With new samples, the Nextera Flex library preparation protocol performed much better. For example, in the next 66 genomes, we got a success rate of whole genome sequencing of 90.9%.

Further comparison of the kits looked at the cost and processing times for each of the methods used. The costs of the library preparation methods, including components not included, such as Ampure bead (Beckman Coulter) and End Repair/dA-Tailing Module (New England BioLabs) kits, varied with a difference of up to 30% (Appendix A). The NEBNext Ultra II DNA library method was found to be the cheaper option. Nextera DNA Flex was found to have the shortest processing time of less than three hours. While all three methods required end repair of amplicons prior to indexing, the Nextera Flex method encompassed the tagmentation step together with the ligation of adapters. There was no need to quantify and normalize individual libraries at the end of library preparation, as normalization occurred during the tagmentation step.

This study has many limitations. Firstly, there was no time to prepare properly for the initial sequencing, as access to the first positive samples was during a large nosocomial outbreak investigation. Secondly, the quality of the samples was not homogeneous, as some samples arrived at our laboratories weeks after being sampled from the patients. Thirdly, unavailability of reagent stocks was common during the lockdown in South Africa, and it was necessary to innovate and adapt the protocols.

To summarize, despite the difficulties posed by the lockdown, the data generation and analysis of a large COVID-19 outbreak in South Africa were completed in just a few weeks. The performance of three library preparation kits were also evaluated for their quality, cost, ease of use, and time efficiency. In addition, a bioinformatics workflow was adapted to assemble SARS-CoV-2 genomes from raw sequence reads in near-real time. All the protocols and raw data used in this study have been made publicly available and distributed to laboratories of the Network for Genomics Surveillance in South Africa (NGS-SA) and the Africa Centre for Diseases Control (Africa CDC).

## Figures and Tables

**Figure 1 genes-11-00949-f001:**
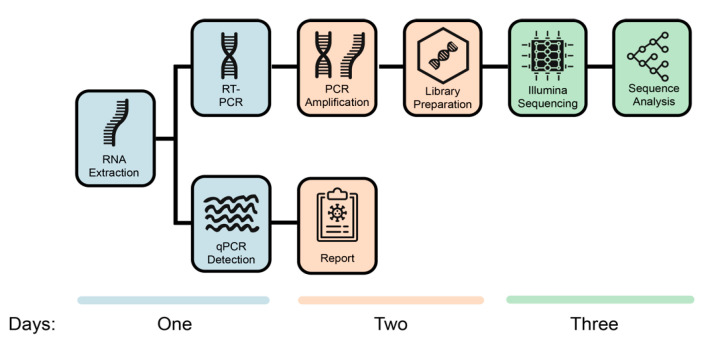
Processes to generate severe acute respiratory syndrome coronavirus 2 (SARS-CoV-2) genomes and qPCR diagnostics in the KwaZulu-Natal Research Innovation and Sequencing Platform (KRISP) laboratory. The figure also shows the number of days needed by two senior scientists to generate 24 whole genomes by using an Illumina Miseq Nano kit V2. It is possible to generate 94 whole genomes with one extra day of sequencing with the use of a MiSeq Reagent Kit v2 (500 cycles).

**Figure 2 genes-11-00949-f002:**
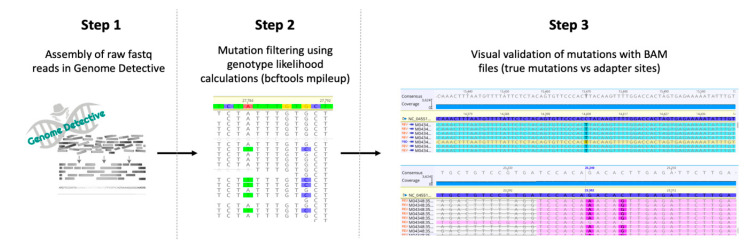
Three-step workflow for generation of high-quality genomes. Step 1: Raw reads from Illumina and Nanopore sequencing were assembled by using the web-based Genome Detective 1.126 (https://www.genomedetective.com/) platform and its coronavirus typing tool. Step 2: The initial assembly obtained from Genome Detective was polished by aligning mapped reads to the references and filtering out mutations with low genotype likelihoods, using bcftools 1.7-2 mpileup method. This calculation determines the probability of a genotype at sites containing reads with various bases (e.g., the probability that position 27,784 is A vs. T in illustration above). Step 3: All mutations were validated visually with BAM files viewed in Geneious software, to ensure that called mutations were true and not part of lingering adapter sites.

**Figure 3 genes-11-00949-f003:**
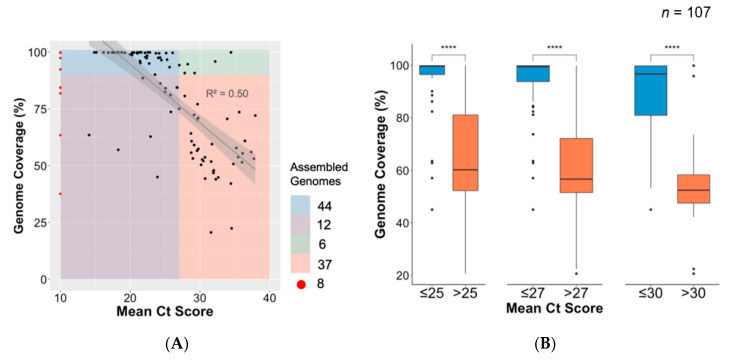
Association between cycle threshold (Ct) value and genome length. (**A**) Regression plot of mean Ct value of all unique samples against their genome lengths (% coverage against SARS-CoV-2 reference). Samples with missing Ct value information (*n* = 8) are shown in red. Forty-four assembled genomes of >90% were produced from samples having Ct value <27 (blue); six genomes of >90% and Ct value >27 (green); 12 genomes <90% coverage and Ct value <27 (purple); and 37 genomes <90% coverage and Ct value >27 (orange). (**B**) Box plot and statistical comparison of genome coverage obtained from samples grouped in three mean Ct value thresholds (25, 27, and 30), showing statistically significant (*t*-tests) differences between lower and higher Ct value samples. ****: level of significance.

**Figure 4 genes-11-00949-f004:**
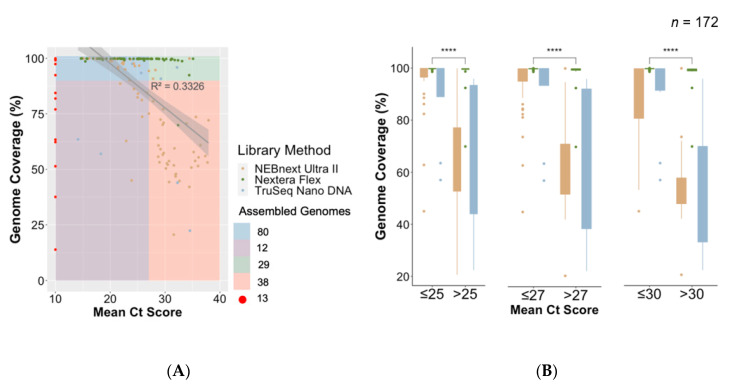
Association between Ct value and genome length by library preparation method. (**A**) Regression plot of mean Ct value of all unique samples against their genome lengths (% coverage against SARS-CoV-2 reference). Samples with missing Ct value information (*n* = 8) are shown in red. A total of 114 assembled genomes of >90% were produced (80 with Ct value <27, 29 with Ct value >27, and five with missing Ct values). (**B**) Box plot and statistical comparison of genome coverage obtained from samples grouped in three mean Ct value thresholds (25, 27, and 30) by library preparation method, showing statistically significant (*t*-tests) differences between lower and higher Ct value samples. ****: level of significance.

**Figure 5 genes-11-00949-f005:**
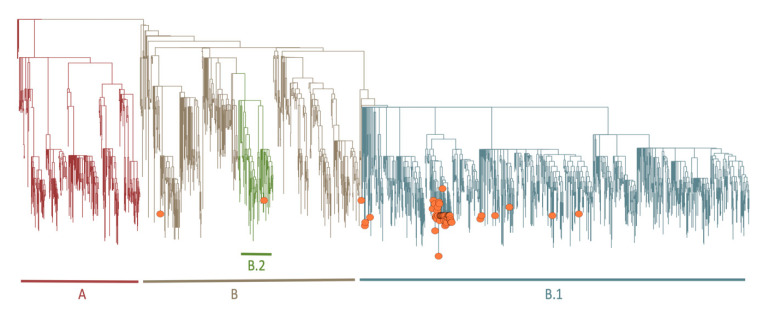
Phylogenetic tree. Showing a Maximum-Likelihood (ML) tree of the 54 genomes (orange circles) against publicly available SARS-CoV-2 genomes as reference. The 54 genomes fall mostly in the B.1 (*n* = 50), B (*n* = 3), or B.2 (*n* = 1) lineages.

**Table 1 genes-11-00949-t001:** Comparison of coverage and Ct values between the different library preparation methods for repeat samples only.

	Coverage (% of SARS-CoV-2 Genome)			
Sequence	TruSeq DNA Nano	NEBnext Ultra II	Nextera Flex	Ct Value
KRISP_0002	97.5	98.3	98.5	24.0
KRISP_0004	99.5	98.3	98.1	24.1
KRISP_0019	97.4	89.9	90.2	NA
KRISP_0021	63.5	99.1	82.7	14.1
KRISP_0024	-	95.2	94.6	21.5
KRISP_0026	-	99.8	99.9	17.9
KRISP_0028	-	96.1	98.1	21.4
KRISP_0031	-	86.2	84.1	24.3
KRISP_016	99.8	-	99.2	NA
KRISP_017	99.9	-	99.9	NA
KRISP_006	99.5	-	96.3	21.0
KRISP_007	99.9	-	99.6	25.5
KRISP_003	25.3	92.4	-	Undetermined
KRISP_010	93.4	92.3	-	25.6
KRISP_014	81.9	70.6	-	NA
KRISP_013	63.4	73.0	-	NA

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

preparation protocols for whole genome sequencing based outbreak investigation. Front. Public Heal..

