# Peer review of "Whole Genome Sequencing of SARS-CoV-2: Adapting Illumina Protocols for Quick and Accurate Outbreak Investigation during a Pandemic"

_genes, 2020, doi:10.3390/genes11080949_

Round 1

Reviewer 1 Report

This article tells an interesting story demonstrating some of the challenges associated with reagent shortages and suboptimal samples, and the resultant quick and skillful improvisation of the staff at the University of KwaZulu-Natal in South Africa to perform whole genome sequencing of SARS-CoV-2.  The workflows and pipelines developed, and the sharing of these with other laboratories in the laboratory network in South Africa, are commendable.  

Whilst the writing tells a sound story, the stylistic structure requires significant improvement before it can be published.  I provide some examples below, however, there are numerous more throughout.  The word “we” and “our” is overused throughout the manuscript and authors should restructure most, if not all, sentences containing “we” and “our” e.g. instead of “we did/used xyz” it should be “xyz was performed/used” etc.  Considerable parts of the Results section (particularly sections 3.2 and 3.5) are more appropriate to go into the Discussion section.  I suggest a thorough proof reading, sentence restructuring and correction of mistakes/inconsistencies (including in the Supplementary Material.)

In the abstract the authors claim to perform the sequencing in “real-time”, which is somewhat misleading.  In the discussion the authors refer to this as “near real-time”, which is a more accurate description. 

In the paper the authors describe the genome coverage in relevance to the Ct values but they do not describe the depth of coverage.  Is there a reason why the depth of coverage analysis was not performed/reported?

Below are some further comments to help improve the manuscript:

Line 28 (abstract): instead of “sequencing in Illumina” I suggest “Illumina sequencing”

Line 77-78 It’s not clear what the addition of “1.25μl RNase P Assay (20X), and 1.25μl of the 2019-nCoV assay (20X)” to the assay actually refers too.  Are these used as controls?  The sentence is not clear and needs to be revised.  It is not possible to “add an assay” to reagents.  Also please pay attention to consistency with upper and lower case.

Line 87: “and or” which is it? Or do the authors mean “and/or”?

Line 88: To whom/where were the results reported?

Headings 2.5, 2.6 and 2.7 should either be subheadings under section 2.4 (i.e. 2.4.1, 2.4.2, 2.4.3) or if this is not possible they should just form unnumbered paragraphs under section 2.4.

Lines 114, 133, 148: Shouldn’t the volume of sodium acetate used be added here?

Line 147: Why is the number “four” spelled out here but not elsewhere?

Line 156: Add version of Geneious software used.

Line 161: “all” should be lower case. 

Line 162: Supplementary Table 6 has not been supplied.  Also correct spelling of “material” in the heading of the supplementary material attachment.

Line 181: “We started sequencing the day after we got involved in the outbreak investigation, on 5 April 181 2020, so there was no time to prepare.” A better way to phrase this might be: “Sequencing commenced one day after (our laboratory insert name) became involved in the outbreak investigation, with no prior notice/insufficient time to prepare.” 

Line 262: Please define KRISP within the body of the article at first occurance. 

Line 166: Add version info for FigTree and ggtree and source info where possible.  Please provide GSAID accession numbers.

Line 167: This sentence seems to suggest that sequences were submitted to a short read archive on GenBank and an accession number was provided, however, this link does not seem to correlate to sequences submitted in this study (it links to a reference genome submitted by someone else.)  Please correct.

Line 177: COVID-19 (the name for the disease) needs to be replaced with SARS-CoV-2 (the virus.)

Line 190: Perhaps this should commence with “Staff at our (or preferably use the name of the lab, instead of "our") laboratory received…” because laboratories can’t receive training (but the staff can.)

Line 260: Should this be “coverage” instead of “length”?

Which currency do the $ values refer to?

Ct might be more appropriately referred to as “values” not “scores” throughout.

Line 315: KZN needs to be defined when used for the first time

Line 321: Instead of the sentence “SARS-CoV-2 sequencing is a rapid and accurate method of analysing genetic variability and identifying transmission chains during outbreaks” a better sentence structure might be: “Whole genome sequencing of SARS-CoV-2 samples has the capacity to provide rapid and accurate information for the analysis of genetic variability and identification of transmission chains during outbreaks.”

Supplementary Table S1: What are the blanks - is it data not available?  If that’s the case then N/A should be entered into the blanks.

Author Response

Question 1: This article tells an interesting story demonstrating some of the challenges associated with reagent shortages and suboptimal samples, and the resultant quick and skillful improvisation of the staff at the University of KwaZulu-Natal in South Africa to perform whole genome sequencing of SARS-CoV-2.  The workflows and pipelines developed, and the sharing of these with other laboratories in the laboratory network in South Africa, are commendable.  

Answer 1: We would like to thank the reviewer for the positive comments and for commending  us on our sharing and open access policies.

Question 2: Whilst the writing tells a sound story, the stylistic structure requires significant improvement before it can be published.  I provide some examples below, however, there are numerous more throughout.  The word “we” and “our” is overused throughout the manuscript and authors should restructure most, if not all, sentences containing “we” and “our” e.g. instead of “we did/used xyz” it should be “xyz was performed/used” etc.  

Answer 2: We have edited the manuscript as suggested to replace the word “we” and “our” with more passive sentences as suggested by the reviewer.

Question 3: Considerable parts of the Results section (particularly sections 3.2 and 3.5) are more appropriate to go into the Discussion section.  I suggest a thorough proof reading, sentence restructuring and correction of mistakes/inconsistencies (including in the Supplementary Material.)

Answer 3: The suggested restructuring was done and thorough proofreading was performed.

Question 4: In the abstract the authors claim to perform the sequencing in “real-time”, which is somewhat misleading.  In the discussion the authors refer to this as “near real-time”, which is a more accurate description.

Answer 4: Changed to “near real-time” in the abstract.

Question 4: In the paper the authors describe the genome coverage in relevance to the Ct values but they do not describe the depth of coverage.  Is there a reason why the depth of coverage analysis was not performed/reported?

Answer 4: It is common to report on the genome coverage and Ct Score (see supplementary info in Gudbjatsson et al. NEJM 2020) . Researchers normally add information on the depth coverage in the GISAID annotation file. That said, we have now presented the depth of coverage in relevance to Ct values in Supplementary figure 2.

Below are some further comments to help improve the manuscript:

Question 5: Line 28 (abstract): instead of “sequencing in Illumina” I suggest “Illumina sequencing”

Answer 5: Line 28 (abstract): “sequencing in Illumina” has been changed to “Illumina sequencing”.

Question 6: Line 77-78 It’s not clear what the addition of “1.25μl RNase P Assay (20X), and 1.25μl of the 2019-nCoV assay (20X)” to the assay actually refers too.  Are these used as controls?  The sentence is not clear and needs to be revised.  It is not possible to “add an assay” to reagents.  Also please pay attention to consistency with upper and lower case.

Answer 6: Thanks for this suggestion, we have edited the text for clarity.

Line 75 (Materials and Methods): “the TaqMan 2019-nCoV assay kit v1 according to manufacturer’s instructions” has been changed to “the TaqMan 2019-nCoV Assay Kit v1 (LifeTechnologies, Frederick, MD, USA) according to the manufacturer’s instructions”.

The 2019-nCoV is a target and RNAse P is a control. For RNAse P, this has been made more clear in Line 86-87 and expressed as “a Ct value for RNAse P was used as an endogenous internal amplification control in each reaction”.

In line 78, the word “assay” has been removed and the sentence reads as “1.25µl RNase P (20X), and 1.25µl of the 2019-nCoV target (20X)”

Question 7: Line 87: “and or” which is it? Or do the authors mean “and/or”?

Answer 7: We mean “and”.

The sentence in Line 86: “To accept the results, we confirmed a Ct value for RNAse P (i.e. an endogenous internal amplification control) and or the target gene in each reaction, with undetermined Ct values in the no template control” was revised to read as “To accept the results, a Ct value for RNAse P was used as an endogenous internal amplification control in each reaction, and the results were only valid if the Ct values in the no template control were undetermined”.

Question 8: Line 88: To whom/where were the results reported?

Answer 8: In Line 88: The sentence “We reported Ct values for each target gene” was removed as the Ct values were generated for the purpose of this paper and not for reporting.

Question 9: Headings 2.5, 2.6 and 2.7 should either be subheadings under section 2.4 (i.e. 2.4.1, 2.4.2, 2.4.3) or if this is not possible they should just form unnumbered paragraphs under section 2.4.

Answer 9: Headings 2.5, 2.6 and 2.7 has been changed to subheadings under section 2.4. The headings now read as follows:

2.4.1. TruSeq Nano DNA Library Preparation

2.4.2. NEBNext Ultra II DNA Library Preparation

2.4.3. Nextera DNA Flex Library Preparation

Heading 2.8. Data analysis has subsequently been changed to 2.5. Data analysis.

Question 10: Lines 114, 133, 148: Shouldn’t the volume of sodium acetate used be added here?

Answer 10: The volume (5 μl) of sodium acetate has been added to lines 114, 133 and 148.

Question 11: Line 147: Why is the number “four” spelled out here but not elsewhere?

Answer 11: This was an error. The sentence “We normalized each sample library to four nM concentration“ has been changed to “Each sample library was normalised to 4nM concentration”.

Question 12: Line 156: Add version of Geneious software used.

Answer 12: Genious version added.

Question 13: Line 161: “all” should be lower case. 

Answer 13: In line 161: “All authors acknowledged in Supplementary Table S6” has been changed to “all authors acknowledged in Supplementary Table S6”.

Question 14: Line 162: Supplementary Table 6 has not been supplied.  Also correct spelling of “material” in the heading of the supplementary material attachment.

Answer 14: This table was attached as an excel file. We have correct the spelling of material.

Question 15: Line 181: “We started sequencing the day after we got involved in the outbreak investigation, on 5 April 181 2020, so there was no time to prepare.” A better way to phrase this might be: “Sequencing commenced one day after (our laboratory insert name) became involved in the outbreak investigation, with no prior notice/insufficient time to prepare.” 

Answer 15: This line was changed to: “Sequencing started the day after our laboratory, KRISP, became involved in the outbreak investigation, with no prior notice or time to prepare.”

Question 16: Line 262: Please define KRISP within the body of the article at first occurrence. 

Answer 16: KRISP was defined in the body of the article.

Question 17: Line 166: Add version info for FigTree and ggtree and source info where possible.  Please provide GSAID accession numbers.

Answer 17: Version of Figtree and ggtree added. GISAID accessions numbers also added.

Question 18: Line 167: This sentence seems to suggest that sequences were submitted to a short read archive on GenBank and an accession number was provided, however, this link does not seem to correlate to sequences submitted in this study (it links to a reference genome submitted by someone else.) 

Answer 18: Apologies for this typo. This has been fixed now in the manuscript.

Question 19: Line 177: COVID-19 (the name for the disease) needs to be replaced with SARS-CoV-2 (the virus.)

Answer 19: COVID-19 was replaced with “SARS-CoV-2 (the virus).

Question 20: Line 260: Should this be “coverage” instead of “length”?

Answer 20: Length was changed to “coverage”.

Question 21: Which currency do the $ values refer to?

Answer 21: “$” was changed to “US$”

Question 22: Ct might be more appropriately referred to as “values” not “scores” throughout.

Answer 22: “Scores” were changed to “values”.

Question 23: Line 315: KZN needs to be defined when used for the first time

Answer 23: KZN has now been defined as KwaZulu-Natal.

Question 24: Line 321: Instead of the sentence “SARS-CoV-2 sequencing is a rapid and accurate method of analysing genetic variability and identifying transmission chains during outbreaks” a better sentence structure might be: “Whole genome sequencing of SARS-CoV-2 samples has the capacity to provide rapid and accurate information for the analysis of genetic variability and identification of transmission chains during outbreaks.”

Answer 24: “SARS-CoV-2 sequencing is a rapid and accurate method of analysing genetic variability and identifying transmission chains during outbreaks” was replaced with: “Whole genome sequencing of SARS-CoV-2 samples has the capacity to provide rapid and accurate information for the analysis of genetic variability and identification of transmission chains during outbreaks.”

Question 25: Supplementary Table S1: What are the blanks - is it data not available?  If that’s the case then N/A should be entered into the blanks.

Answer 25: Fixed.

Reviewer 2 Report

This is a nicely written paper describing the workflow and hurdles of a lab trying to rapidly sequence SARS-CoV-2 Virus.

While the Story is impressive, with the lab providing good results in the face of adversity, and it will be of some interest to those performing the same tasks, I imagine that most labs have already been through these steps, and if not they will be sending samples to larger sequencing centres.

Some specific comments:

Line 51 how were These Patients and their travel Backgrounds identified?

Line 63 please describe here that These are remnants from various diagnostic labs. This is clarified later, but belongs here. If method of diagnosis is known, also add here.

Line 73, please move storage to section above.

Line 122 and elsewhere: what is stockout? Unavailability of stock?

Line 159 spelling Rambaut

Line 248, the increase here is not significatn at p<0.05. Please rephrase.

Spelling of Geneious in Suppl Fig 1

I see more often "Ct value" than "Ct score" Please check.

Why are CT values given as 5 Digit numbers? I see them as 2 Digit values with 1 dp usually.

Line 261, Higher coverage is associated with lower Ct score, not an increase.

Line 282 16 samples

Line 286: These  % and numbers don't match.
Line 375 title?
Line 376 identifies?
Table 3 is missing some GISAID IDs
The details on phone calls etc is likely outside the scope of the publication.
Author contributons, Funding, Acknowledgements and COnflicts of interests are all missing.

Comments on data Analysis:
I am confused about how this works in the software mentioned. The figures indicate that this is mapping to a reference (should be given in methods), but the text states that it is assembly. If assembly, then #contigs, assembly length, and other parameters would be interesting. This needs to be clarified.

Trimming of reads prior to analysis should have removed adapters. The authors do not state clearly whether or not adapters or primer sequences were actively removed or not. This can likely impair the discovery of SNPs in the primer binding sites, making the methods more prone to false negatives.
Why was this not performed, and would this not improve results and simplify workflow?

Figure 1: how was Miseq nano 500 performed in under 1 day. As far as I am Aware this step alone takes 28h. This makes me question the Turnaround time.

Please state in methods / line 245 that after visual checking, the sequences were manually edited as required. THis is not clear.
The authors refer to visual validation of mutation. This process, although commendable as a reality check, cannot be the only filter deployed to determine the variant call. The authors should rather provide and justify a set of numerical (and thereby reproducible) thresholds used to discriminate ambiguous from accepted variant calls.

How did the steps increase coverage? This is not intuitive.
Figure 3: The authors refer to a correlation between Genome coverage and mean Ct value, which is consistent to previous literature in the filed. However, prior to the deployment of regression (it seems linear regression from the plot, it is however not specified by the authors) the authors should ascertain whether the data fit a linear or not linear model.
On top of this, the author do not provide a correlation coefficient, which would be essential to estimate the goodness of fit.

Author Response

Question 26: This is a nicely written paper describing the workflow and hurdles of a lab trying to rapidly sequence SARS-CoV-2 Virus.

Answer 26: We would like to thank the reviewer for the positive comment.

Question 27: While the Story is impressive, with the lab providing good results in the face of adversity, and it will be of some interest to those performing the same tasks, I imagine that most labs have already been through these steps, and if not they will be sending samples to larger sequencing centres.

Answer 27: Thanks for this comment and we agree with the reviewer that this may be happening in the most developed countries in the world that have large sequencing centers. However, this is not the case in most African or Latin America countries. For example, before our protocol and pre-print, we were the only laboratory doing large scale genotyping of SARS-CoV-2 in South Africa. At the time of writing this response, African laboratories have produced 954 whole viral genomes, (which) 296 of which were from South Africa and 235 from our laboratory.

Some specific comments:

Question 28: Line 51 how were These Patients and their travel Backgrounds identified?

Answer 28: The travel backgrounds were identified and only one patient had a travel history as the majority of the cases were from a nosocomial outbreak.

Question 29: Line 63 please describe here that These are remnants from various diagnostic labs. This is clarified later, but belongs here. If method of diagnosis is known, also add here.

Answer 29: Line 63 was revised to : “Remnant samples from various diagnostic laboratories were obtained. These  were collected using nasopharyngeal and oropharyngeal swabs from qPCR confirmed, COVID-19 patients”.

Question 30: Line 73, please move storage to section above.

Answer 30: Storage was removed from this line.

Question 31: Line 122 and elsewhere: what is stockout? Unavailability of stock?

Answer 31: “Stock out” was changed to “unavailability of stock/ reagents”

Question 32: Line 159 spelling Rambaut

Answer 32: “Rambault et al., [18]” has been changed to “Rambaut et al., [18]”

Question 33: Line 248, the increase here is not significatn at p<0.05. Please rephrase.

Answer 33: The difference between matches between step 2 and step 3 is indeed significant. Sentence changes to the following to make it clearer: “The quality of  the consensus sequences clearly increased, as indicated by significantly higher (t-test) coverage, concordance, identity and matches at Step 2 compared to Step 1, and significantly higher  matches at Step 3 compared to Step 2 (after visual validation) (Supplementary Figure S1, Table S2)”.

Question 34: Spelling of Geneious in Suppl Fig 1

Answer 34: Fixed

Question 35: I see more often "Ct value" than "Ct score" Please check.

Answer 35: CT “score” has been changed to Ct “value”

Question 36: Why are CT values given as 5 Digit numbers? I see them as 2 Digit values with 1 dp usually.

Answer 36: CT values were changed to 2 digit values with one decimal place.

Question 37: Line 261, Higher coverage is associated with lower Ct score, not an increase.  

Answer 37: Fixed

Question 38: Line 282 16 samples

Answer 38: Fixed

Question 39: Line 286: These  % and numbers don't match.

Answer 39: Fixed

Question 40: Line 375 title?

Answer 40: Fixed

Question 41: Line 376 identifies?

Answer 41: Fixed

Question 42: Table 3 is missing some GISAID IDs

Answer 42: Fixed

Question 43: The details on phone calls etc is likely outside the scope of the publication.
Author contributions, Funding, Acknowledgements and Conflicts of interests are all missing.

Answer 43: Details on author contributions, funding, acknowledgements and conflict of interest were added to the acknowledgement section of the manuscript.

Question 44: I am confused about how this works in the software mentioned. The figures indicate that this is mapping to a reference (should be given in methods), but the text states that it is assembly. If assembly, then #contigs, assembly length, and other parameters would be interesting. This needs to be clarified.

Trimming of reads prior to analysis should have removed adapters. The authors do not state clearly whether or not adapters or primer sequences were actively removed or not. This can likely impair the discovery of SNPs in the primer binding sites, making the methods more prone to false negatives.
Why was this not performed, and would this not improve results and simplify workflow?

Answer 44: We apologize for the confusion. All these processed are carried out automatically by our software Genome Detective. We have added the following description to Data Analysis methods to make this clearer:

Genome Detective pre-processes sequencing reads with Trimmomatic. Reads are then filtered and sorted on a protein level into “bins” corresponding to recognized viral species using DIAMOND and the UniRef90 database. Reads in each bin are assembled against the corresponding viral reference using SPADes assembler. Genome Detective produces consensus sequences from the assembled contigs for the identified viruses. In our SARS-CoV-2 assembly workflow. The initial assembly obtained from Genome Detective was polished by aligning mapped reads to the reference (NC_045512) and filtering out mutations with low genotype likelihoods using bcftools 1.7-2 mpileup method.”

Question 45: Figure 1: how was Miseq nano 500 performed in under 1 day. As far as I am Aware this step alone takes 28h. This makes me question the Turnaround time.

Answer 45: The extraction, CDNA synthesis and PCR amplification are done on Day 1, with amplification left on the thermal cycler overnight. Library preparation takes place on Day 2 and the libraries are loaded onto the sequencer on Day 2. The Miseq Nano run is usually 24hrs. The sequencing run is complete on Day 3 and the results are analyzed on Day 3. We also added information about the use of a larger sequencing kit in Figure 1, the  MiSeq Reagent Kit v2 (500-cycles), which takes two days (i.e. 48h) to complete sequencing but can geneate up to 94 genomes per run.

Question 46: Please state in methods / line 245 that after visual checking, the sequences were manually edited as required. This is not clear.

Answer 46: Added

Question 47: The authors refer to visual validation of mutation. This process, although commendable as a reality check, cannot be the only filter deployed to determine the variant call. The authors should rather provide and justify a set of numerical (and thereby reproducible) thresholds used to discriminate ambiguous from accepted variant calls.

Answer 47: Thanks for this suggestion. We normally look at the minority variants. If a variant is at prevalence of more than 80%, it is selected as the accepted variant.  

Question 48: How did the steps increase coverage? This is not intuitive.

Answer 48: The second step increased coverage because we are now mapping reads to a reference sequence using mpileup and bcftools, while in the first step it was a de-novo assembly using SPADES. De-novo assembly will reject more reads than a reference based assembly.

Question 49: Figure 3: The authors refer to a correlation between Genome coverage and mean Ct value, which is consistent to previous literature in the filed. However, prior to the deployment of regression (it seems linear regression from the plot, it is however not specified by the authors) the authors should ascertain whether the data fit a linear or not linear model. -
On top of this, the author do not provide a correlation coefficient, which would be essential to estimate the goodness of fit.

Answer 49: Edited to the following sentence: “There was a marked linear association between higher coverage and samples with a lower Ct value (R2 = 0.50) (Figure 3).”